# A Collagen Basketweave from the Giant Squid Mantle as a Robust Scaffold for Tissue Engineering

**DOI:** 10.3390/md19120679

**Published:** 2021-11-29

**Authors:** Anastasia Frolova, Nadezhda Aksenova, Ivan Novikov, Aitsana Maslakova, Elvira Gafarova, Yuri Efremov, Polina Bikmulina, Vadim Elagin, Elena Istranova, Alexandr Kurkov, Anatoly Shekhter, Svetlana Kotova, Elena Zagaynova, Peter Timashev

**Affiliations:** 1World-Class Research Center “Digital Biodesign and Personalized Healthcare”, Sechenov First Moscow State Medical University (Sechenov University), 8-2 Trubetskaya Street, 119991 Moscow, Russia; gafarova_e_r@staff.sechenov.ru (E.G.); efremov_yu_m@staff.sechenov.ru (Y.E.); bikmulina_p_yu@staff.sechenov.ru (P.B.); timashev_p_s@staff.sechenov.ru (P.T.); 2Institute for Regenerative Medicine, Sechenov First Moscow State Medical University (Sechenov University), 8-2 Trubetskaya Street, 119991 Moscow, Russia; aksenova_n_a@staff.sechenov.ru (N.A.); istranova_e_v@staff.sechenov.ru (E.I.); kurkov_a_v@staff.sechenov.ru (A.K.); shekhter_a_b@staff.sechenov.ru (A.S.); kotova_s_l@staff.sechenov.ru (S.K.); 3N.N. Semenov Federal Research Center for Chemical Physics, RAS, 4 Kosygin Street, 119991 Moscow, Russia; 4Research Institute of Eye Diseases, 11 Rossolimo Street, 119021 Moscow, Russia; i.novikov@niigb.ru; 5Faculty of Biology, Department of Human and Animal Physiology, M.V. Lomonosov Moscow State University, 1-12 Leninskie Gory, 119991 Moscow, Russia; aitsana.dokrunova@gmail.com; 6Institute of Experimental Oncology and Biomedical Technologies, Privolzhsky Research Medical University, Minin and Pozharsky Square 10/1, 603950 Nizhny Novgorod, Russia; elagin.vadim@gmail.com; 7Institute of Experimental Oncology and Biomedical Technologies, National Research Lobachevsky State University of Nizhny Novgorod, Prospekt Gagarina (Gagarin Avenue) 23, 603950 Nizhny Novgorod, Russia; ezagaynova@gmail.com; 8Chemistry Department, M.V. Lomonosov Moscow State University, 1 Leninskie Gory, 119991 Moscow, Russia

**Keywords:** biomechanical properties, collagen membrane, AFM, SEM, tensile test, giant squid, Dosidicus Gigas, jumbo squid, outer tunic, tissue engineering

## Abstract

The growing applications of tissue engineering technologies warrant the search and development of biocompatible materials with an appropriate strength and elastic moduli. Here, we have extensively studied a collagenous membrane (GSCM) separated from the mantle of the Giant squid Dosidicus Gigas in order to test its potential applicability in regenerative medicine. To establish the composition and structure of the studied material, we analyzed the GSCM by a variety of techniques, including amino acid analysis, SDS-PAGE, and FTIR. It has been shown that collagen is a main component of the GSCM. The morphology study by different microscopic techniques from nano- to microscale revealed a peculiar packing of collagen fibers forming laminae oriented at 60–90 degrees in respect to each other, which, in turn, formed layers with the thickness of several microns (a basketweave motif). The macro- and micromechanical studies showed high values of the Young’s modulus and tensile strength. No significant cytotoxicity of the studied material was found by the cytotoxicity assay. Thus, the GSCM consists of a reinforced collagen network, has high mechanical characteristics, and is non-toxic, which makes it a good candidate for the creation of a scaffold material for tissue engineering.

## 1. Introduction

The basic objective of tissue engineering consists of obtaining such a scaffold material that would promote complete or at least partial regeneration of internal organs, skin, vascular, bone, cartilage, and other tissues. For a construct to successfully engraft in the body, its parameters ought to be similar to those of the region in which the construct will function. The construct’s biocompatibility, biodegradability, as well as mechanical properties determine its potential to substitute the corresponding live tissue in the body.

The interest to collagen-based materials is stipulated by the fact that collagen is biocompatible with the recipients’ tissues, can biodegrade, is non-toxic, non-carcinogenic and non-immunogenic, and combines many characteristics of synthetic polymers (strength, stiffness, ability to form various supramolecular structures, etc.).

Currently, a plethora of pharmaceutical preparations and medical devices have been created using collagen as a base [1].

Collagen is one of basic natural materials which have application in tissue engineering [2,3,4,5,6]. In many types of connective tissue, it is a fibrillar protein and the main component responsible for the tissue integrity, shape, elasticity, and strength.

Connective tissue is present in all the organs and tissues in the body and comprises ~60–90% of their weight. Such collagenous structures as tendons [7,8] and ligaments [9] have a certain structural hierarchy of collagen to withstand intensive mechanical loads (extension and compression). The common idea about them has been that tendons and ligaments are structurally similar, if not identical [10]. Ligaments [10,11,12] and tendons [7,10,13] consist of tightly packed parallel collagen fibers. Ligaments differ from tendons by the predominance of elastic fibers; therefore, they are characterized by a lower strength but higher flexibility as compared to tendons. The distinctions between them stem from the connections they create; ligaments connect a bone with another bone, and tendons connect a muscle with a bone. A number of studies are dedicated to the restoration of tendon ruptures using different materials [14,15].

The skin is also an interesting and sophisticated collagen-based organ [16]. The skin structure resembles a net consisting of differently oriented collagen fibers [17,18,19]. One of the basic functions of the skin is to protect internal organs and tissues from mechanical injuries. Skin as a material exhibits a viscoelastic behavior, and its mechanical response to a stress involves both a viscous component related to energy dissipation and an elastic component related to energy storage [20,21]. Collagen fibers comprise 75% of the skin tissue dry weight [22], and it is those fibers that are responsible for the skin strength. The skin’s mechanical properties are important for a number of applications, including surgery, dermatology, forensic medicine, etc. [23]. The problem of skin replacement and search of the appropriate materials has long been discussed (see, for example, a review [24]).

The anulus fibrosus, an outer fibrous ring of the intervertebral disc, is yet another example of a collagen-based tissue that undergoes mechanical stresses of various directions and has a corresponding collagen packing [25,26].

The knowledge of the mechanical characteristics of tissues and organs, and the conditions in which they must function, help to create or select a material that would be appropriate for their complete or partial replacement [27,28]. The basic mechanical properties include the strength, stiffness, viscosity, elasticity, plasticity, brittleness, etc.

Table 1 presents examples of tissues with a certain collagen packing related to their mechanical properties and materials for tissue engineered constructs meant to replace such tissues.

In the view of creating tissue engineered constructs with predefined mechanical properties, the mantle of squids attracts special attention, since these animals, living in the deep under great pressure, must have a robust musculature and outer coating to protect their internal organs. The morphological features of the squids’ mantle affect the mechanisms of their locomotion [46].

Among many species of cephalopods, the Giant squid (Humboldt squid) represents the most important object of fishing, which covers 30% of the world fishing volume and about 4% of the entire world market of squids [47]. It is the biggest of the known mollusks. This species of large predatory squids lives in the eastern Pacific region, along the Peru coast at depths of 200 to 700 m. Its mantle can reach almost two meters in length, while its lifespan is only about 2 years, since the squid dies upon spawning [48].

A number of publications have appeared recently on the use of marine collagen, obtained, for example, from fish scales [49], mantle, fins, and tentacles of squids [50], and even sea cucumbers [51]. Uriarte-Montoya et al. (2010) described a film for application in the food and medical industries, prepared from collagen extracted from the mantle of the Giant squid of the *Dosidicus gigas* species [43]. Adamowicz et al. (2021) conducted a study on the use of the decellularized mantle of *Loligo vulgaris* squid in tissue engineering as a material for the urethra reconstruction [52]. Collagen-based materials prepared from the mantle of the Giant squid might also become a prospective carrier in tissue engineering, however, no studies on this idea have been reported so far.

Oliveira et al. (2021) discuss the application directions and advantages of marine collagen, as well as the need for the research in this area, aiming at strengthening this biopolymer’s position on the world’s collagen market [53]. Physically, biochemically, and spectroscopically, marine collagen is identical to mammalian collagen [54,55].

Application of mollusks for collagen production has other advantages, including safety from Creutzfeldt–Jakob disease, which is associated with collagen obtained from cattle, and no ethical or religious barriers.

In this study, our objective was to assess the possibility of using a material obtained from the mantle of the Giant squid, *Dosidicus gigas*, for the tasks of regenerative medicine, based on the data on its chemical composition, structural analysis, biomechanical properties, and cytotoxicity. The studied material represented a collagenous membrane prepared from the squid’s outer tunic (hereafter, Giant Squid Collagenous Membrane, GSCM).

## 2. Results

### 2.1. Collagen Is a Basic Component of the GSCM

#### 2.1.1. Amino Acid Analysis

According to the amino acid analysis, the content of hydroxyproline (Hyp) in the GSCM was 86.3 residues, proline (Pro)—91.3 residues per 1000 residues (Table 2). The Hyp percentage in the studied specimen was 10.13 weight %.

The specimen also contained a large amount (330 per 1000 residues) of glycine (Gly). The weight percentage of Gly was 22.18%. This finding is related to the fact that a molecule of collagen consists of a triple helix formed by three polypeptide helical strands, and each helical chain is formed by three-residue-long repeats, with glycine as one of the three residues. Thus, the primary structure of collagen is characterized by a large content of glycine. The high content of glutamic acid (Glu) in the specimen is explained by the presence of proline (Pro) since Pro is synthesized from glutamic acid.

#### 2.1.2. SDS-PAGE

The SDS-PAGE analysis showed four main bands in the studied GSCM (Figure 1). Two bands had the molecular weights of 133.3 and 151.6 kDa, and they were assigned to two α-chains of collagen, α1 and α2. The two high-molecular components, with the weights of 295.7 kDa and 300 kDa, were identified as a β-chain consisting of two α-chains and a γ-chain consisting of three α-chains, respectively.

### 2.2. Hydration and Thermal Properties of the GSCM

#### 2.2.1. FTIR Spectroscopy

The IR spectrum of the GSCM (blue curve in Figure 2) shows bands at 876, 918, 939, 972, 1030, 1060, 1080, and 1119 cm^−1^, which are characteristic of carbohydrate moieties (CO stretching and COC stretching); an Amide III band at 1236 cm^−1^ (associated with CN stretching and NH deformation); bands positioned at 1336 and 1451 cm^−1^, attributable to methylene vibrations (CH_2_ deformation and CH_3_ deformation); N–H in-plane bend and the C–N stretching vibrations at 1540 cm^−1^ (Amid II). The polypeptide backbone CO stretching vibration is found in the range of 1600–1700 cm^−1^: bands at 1740 cm^−1^ due to carbonyl vibrations, and the one 1630 cm^−1^ due to Amide I. The spectrum shows bands at 2878 and 2927 cm^−1^ assigned to aliphatic chains (CH stretching and CH_3_ stretching) an Amide B band at 3073 cm^−1^ (NH stretching), and a broad band at 3500–3300 cm^−1^ related to Amide A (NH stretching) and OH vibrations [56,57,58,59,60].

For comparison, the FTIR spectra of collagens Type I and Type II were examined [61]. The spectra of both samples are presented in Figure 2, and the band positions are presented in Appendix A. FTIR confirmed a similar triple helical structure with the secondary α-chain structure for all three samples [56]. The IR spectra of the GSCM and collagen type II differed slightly in regard to the bands at 1740, 2800–2930, and 3620–3690 cm^−1^ (associated with OH stretching and H-bonding). From the general view of the spectra, one can assume that the GSCM belongs to collagen Type II, but the increased intensity of bands at 2930 and 1740 cm^−1^ indicates that it rather belongs to a mixture of collagens Type I and Type II. These results confirm the results of SDS-PAGE (see Section 2.1, Figure 1).

The position of the Amide I band in the GSCM spectrum is in agreement with the literature data on the Amide I band in the spectra of oligopeptides containing Gly, Pro, and Ala in various combinations, as well as the spectra of polyproline [60]. This is consistent with the results of the study of the amino acid composition, demonstrating that the main amino acids of the GSCM are Gly, Pro, Hyp, and Glu (see Section 2.1, Table 2).

#### 2.2.2. TGA/DSC Studies

A typical weight loss vs. temperature curve (a thermogravimetric, TG curve), as well as a DSC curve, for the GSCM are displayed in Figure 3. In TG curves, there were two temperatures at which the onset of the thermal degradation occurred. The DSC curves showed two endothermic peaks. The broad endothermic peak in DSC curves in the temperature range of 50–170 °C is associated with thermal dehydration [62,63,64]. This process was accompanied by a ~10% weight loss (the TG curve). The broad and multimodal endothermic peak in the temperature range of 220–330 °C is assigned to the collagen matrix thermal denaturation and destruction. The latter process was accompanied by a ~65% weight loss (TG curves). According to [64], for dehydrated collagen type I, the endothermic peak of denaturation was observed at Tdn = 225 °C. It can be assumed that, below the temperature of Tdn ~225–235 °C, the interchain hydrogen bonds rupture, dehydrated collagens unfold, and amorphous polymers form. The second stage of destruction was observed at Tdst ˃ 235 °C. In general, the GSCM TG and DSC curves were similar to those for collagenous materials [62,64].

#### 2.2.3. Shrinkage Temperature

The shrinkage temperature of the GSCM was experimentally found at 58 °C. The swelling degree was measured as 102% in distilled water and as 176% in PBS. The much higher degree of swelling in the PBS medium is due to the fact that ions present in the saline facilitate hydration of collagen fibers.

### 2.3. Morphological Properties of the GSCM

#### 2.3.1. Histological Studies

The histological studies of the GSCM cross-sections showed that the material had a layered structure that consisted of 8–12 tightly packed “laminae” with the total thickness of ~50–70 µm (Figure 4(A1,B1,C1,D1)). The thickness of each lamina was ~5–7 µm. When stained with hematoxylin and eosin, the material of laminae had uniform eosinophilic staining (Figure 4(A1)). However, the picrosirius red stain (Figure 4(B1)), especially, when using phase contrast (Figure 4(C1)) and polarized light microscopy (Figure 4(D1)) showed that in some regions the material had a fibrillar structure due to poorly visible small collagen fibers oriented along the laminae. In the polarized light microscopy images, these fibers produced a bright glow in the material, testifying the birefringence (anisotropy) specific for oriented fibers in collagen.

#### 2.3.2. Scanning Electronic Microscopy Studies (SEM)

The SEM studies revealed that the GSCM surface had a multilayered basketweave structure, with laminae laid at different angles, which resembled a reinforcing mesh. The angle between the laminae was ~60–90° (Figure 5a,b).

The reinforcing layers consisting of laminae have a definite mutual layer-by-layer orientation. Each layer represents a set of parallel laminae with the width of 38–50 µm and thickness of 4.0–4.5 µm. In turn, each lamina consists of tightly packed parallel collagen fibers longitudinally packed along the whole lamina length (Figure 6a,b). Besides, there is a thin layer that covers the upper reinforcing layer with laminae (Figure 6c). This surface is extremely stable chemically (it was not damaged by the sample preparation procedure) and is formed by a randomly crossed motif of collagen fibers and fibrils.

We also studied the GSCM cross-section using SEM, which showed the layered structure, in agreement with the histological data. The SEM images (Figure 7a,b) demonstrate that laminae change their angle in each layer, thus making a basketweave multilayered collagen structure.

#### 2.3.3. Laser Scanning Microscopy (LSM) (Second Harmonics Generation Signal—SHG)

The LSM studies revealed the SHG signal from collagen Type I and Type II in the sample. In consistency with SEM, it was found that collagen in the GSCM was bundled into laminae with the width of about 60 µm. Laminae located at different depths have different, up to perpendicular, mutual orientation (the angle of packing is ~60–90°). Laminae consist of longitudinally positioned parallel collagen fibers (Figure 8a). At the surface of some regions, bundles of collagen in the form of cords are found (Figure 8b). Similar structures were observed by SEM, as well (Figure 7a,c).

#### 2.3.4. Atomic-Force Microscopy (AFM)

The microrelief of the GSCM surface was visualized using AFM. As seen from Figure 9, the GSCM surface has a fibrillar structure, with collagen fibers consisting of tightly packed longitudinally oriented fibrils.

For comparison, we obtained the topography of the outer tunic of another squid species, *B. magister*, which has an essentially smaller size. As seen from Figure 10, the collagen structure of the outer membrane of this squid species is similar to that of the GSCM, with the corresponding scaling. The basketweave structure of both squids’ reinforcing layers in the outer tunic is clearly visible in AFM images, which testifies the universal character of this structure. Since laminae comprising the reinforcing layer in the GSCM are rather wide (40–50 µm) and located at a certain angle relative to each other, AFM cannot visualize the whole laminar motif of the GSCM, even at the largest available scan size, 100 × 100 µm, so only one cell of the basketweave is seen (Figure 9a). However, for the small squid, *B. magister*, this laminar motif is clearly visible at a 50 × 50 µm scan (Figure 10a), since the *B. magister* has the proportionally smaller mantle and outer tunic thickness (Table 3).

With a higher resolution (a 3 × 3 µm scan, see more on the Figure 11), one can see the characteristic striation of collagen fibrils (D-period). The D-period is equal to 67 nm, although the experimentally obtained values depend on the sample hydration [65].

**Figure 11 marinedrugs-19-00679-f011:**
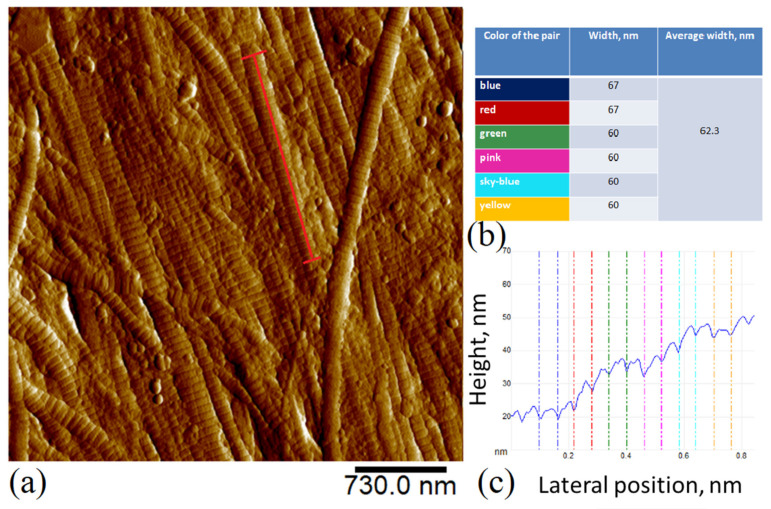
Molecular packing of collagen in the GSCM (**a**) AFM topography (Peak Force Error channel), scan size is 3 × 3 µm; (**b**,**c**) D period of an individual fibril longitudinal section (red line on the topography image). The section shows the characteristic D-period of collagen ([66,67]).

**Table 3 marinedrugs-19-00679-t003:** Mechanical properties of the GCSM and collagenous membranes from other squid species.

Type of Squid	DML, cm	T,µ m	W,µ m	E(w), MPa	UTS(w), MPa	Max ε(w), %	E(d), GPa	UTS(d), MPa	Max ε(d), %	E(w), MPa
				Macromechanical Properties						Micromechanical Properties
Dosidigus gigas	1500–2000	50–70	40–50	20 ± 6	20 ± 8	47 ± 9	1.5 ± 0.5	80 ± 20	20 ± 15	4.1 ± 0.5
Loligo peale [68]	30–50	20–35	2–7	No data	No data	No data	No data	No data	No data	No data
Berryteuthis magister	25	20	4–7	54 ± 17	10 ± 3	27 ± 7	0.4 ± 0.2	28 ± 9	16 ± 5	6.5 ± 0.5

DML—dorsal mantle length of squid; T—thickness of GSCM; W—width of lamina GSCM; E(w)—Young’s modulus of wet GSCM; UTS(w)—ultimate tensile strength of wet GSCM; Max ε(w)—maximum elongation of wet GSCM; E(d)—Young’s modulus of dry GSCM; UTS(d)—ultimate tensile strength of dry GSCM; Max ε(d)—maximum elongation of dry GSCM.

### 2.4. Mechanical Properties of the GSCM

#### 2.4.1. Uniaxial Stretching Tests

The uniaxial stretching tests with the final sample rupture showed that the GSCM of *D. gigas* contained at least two basic directions of collagen fibers (Figure 12). The selected directions of collagen bundles may lead to the complex dependency of the GSCM mechanical properties on the deformation direction.

As seen from the results presented in Table 3, the studied samples of the GSCM of the *D. gigas* species had a rather high tensile strength for a biological material. The Young’s modulus of a dry sample was 1.5 ± 0.5 GPa, while, after 20-min-long hydration of the material, its Young’s modulus drastically dropped to 20 ± 6 MPa. The ultimate tensile strength of the hydrated sample also essentially decreased, however, the strain at rupture grew (to almost 50%).

For comparison, we tested the collagenous membrane of the *B. magister* squid, since it has a similar structure, as shown by AFM. The *B. magister* membrane demonstrated similar mechanical properties as well. Its Young’s modulus was somewhat higher than that of the GSCM, while the ultimate tensile strength and maximum elongation at rupture were slightly lower (in the hydrated state). However, in general, the membrane from the *B. magister* squid is more deformable due to its lower thickness (20 µm).

#### 2.4.2. Micromechanical Properties Studied by AFM

As a result of the AFM-based nanoindentation studies at the micro- and nanoscale, the Young’s modulus of the GSCM surface was measured as 4.1 ± 0.5 MPa. The corresponding value for the *B. magister* squid was slightly higher, 6.1 ± 0.5 MPa. The observed difference between the values at the macro- and microscale is related to the different packing and thickness of collagen structures at different levels. However, the values belong to the same order of magnitude.

### 2.5. Cytotoxicity and Biodegradability of the GSCM

#### 2.5.1. Viability Test

To assess the potential GSCM cytotoxicity, cell viability and proliferation assays were performed. The MSC primary culture was chosen because MSCs are commonly applied in tissue engineering [69,70,71] and were shown to be more sensitive to toxic agents than 3T3 or L929 cell lines [72,73,74]. MSCs seeded at a concentration of 5000 cells per well and exposed to the GSCM extracts at any dilution showed neither reduction in the cell viability nor a decrease in the proliferation rate (Figure 13A). In contrast, both of the assays showed a significant drop (to 20% of the cell viability compared to the control cells) in the cell viability in the presence of SDS at a concentration of 0.05 mg/mL and higher (Figure 13B). Hence, the GSCM does not contain any cytotoxic compounds that could be released during cultivation. The adhesive properties of the GSCM were also shown to be appropriate—MSCs successfully adhered to GSCM films, remained viable during 3 days of cultivation, and proliferated on them. The metabolic activity of cells cultured on the surface of the GSCM was slightly higher than that of the monolayer control (Figure 13C). However, proliferation of collagen-cultivated cells was inhibited in comparison to the monolayer cell culture grown on culture plastic, probably due to the different mechanical properties of the surface. The Live/Dead assay of the GSCM revealed normal MSC spindle-shaped morphology and outnumbering living cells relative to the dead ones (Figure 13D–G). Overall, despite the decreased proliferation rates of cells, the GSCM was shown to maintain the normal cell metabolic activity, proliferation capacity, and morphology both by the extraction and contact cytotoxicity test.

#### 2.5.2. Resistance to Collagenase

The sensitivity to collagenase was studied in order to estimate the biodegradability of the GSCM. The collagenase cleavage study showed that in 6 h the GSCM was digested by 85 ± 5% from the initial weight.

The histological study of the GSCM samples treated with collagenase showed signs of their destruction in the form of the loss of the typical structure, as well as changes in the tinctorial and optical properties of the laminar material (see Section 2.3, Figure 4(A2,B2,C2,D2)). These signs included homogenization with lysis and appearing basophilic (Figure 4(A2)) and picrinophilic regions (Figure 4(B2)), as well as loosening and loss of orientation of collagen fibers (Figure 4(C2)) with the disappeared anisotropy (Figure 4(D2)). At the same time, in the picrosirius red-stained samples, the remaining material represented a homogenic picrinophilic mass, in which few chaotically located destroyed collagen fibers were seen.

#### 2.5.3. LAL Test

To further assess the GSCM biosafety, we tested its pyrogenicity. The most common pyrogens are endotoxins derived from the cell walls of gram-negative bacteria. The LAL test is commonly applied to assess their concentration and is one of the two assays recognized by the U.S. Pharmacopeia (USP) for medical devices. For the GSCM extract, we revealed that the endotoxin level was 0.28 EU/mL, which does not exceed the concentration permitted (0.5 EU/mL) [75,76]. Therefore, the GSCM did not contain endotoxins able to induce a notable pyrogenic reaction. We also performed preliminary in vivo testing of GSCM samples implanted in rats (see Appendix A). It showed that the intact GSCM was still poorly compatible with the host tissues and caused notable inflammatory reaction. However, the GSCM treatment with supercritical carbon dioxide before implantation solved this problem, reducing the inflammatory reaction to only insignificant.

## 3. Discussion

The results of the biochemical and structural studies confirm that collagen is a basic component of the GSCM material. The amino acid analysis showed a high content of Hyp, which is known as a detector for the presence of collagen [77]. Its weight percentage in the samples was 10.13%, while the content of Pro and Hyp in the extracted collagen from the GSCM was 10.9% and 2.8%, respectively [43]. The presence of Cys might indicate that the GSCM possibly contains traces of elastin [78]. Gauza-Włodarczyk et al. (2017) found a similar amino acid composition for bone collagen in [79].

The comparative SDS-PAGE analysis of the GSCM with collagen Type I and Type II revealed the similarity of the GSCM collagen to collagen Type I, based on the characteristic bands. Nam et al. (2008), in the study [80], described collagen extracted from a squid’s skin and compared its physicochemical properties with those of collagen prepared from bovine tendons. The similarity between the two was found, and the squid collagen was classified as Type I.

FTIR demonstrated the presence of collagen Type II, also, in the GSCM. The DSC study showed that the GSCM collagen behaved similarly to both collagen types. The characteristic shrinkage temperature also confirmed the collagenous nature of the GSCM.

The extensive morphology study, including histology, SEM, LSM, and AFM, showed the presence of ordered collagen structures at various levels of organization. From the ultrastructure of fibrils to fibers and fiber bundles, they are characterized by tight packing, orientation, and formation of a basketweave from larger collagen units, laminae. Such a sophisticated arrangement of collagen structures is apparently related to the mechanical properties of the GSCM, such as high strength and Young’s modulus.

Based on the SEM study, we have deduced a possible concept of the collagen arrangement in the studied material, displayed in Figure 14. The arrows in Figure 14 indicate which SEM-revealed feature corresponds to each component of the schematic structure. The structure and packing of laminae revealed by SEM are confirmed by the other structural techniques.

We have not found any published studies on the structure of the collagenous membrane from the Giant squid of the *D. gigas* species, based either on SEM or on any other visualization technique. However, the squid mantle is known to consist of three layers: muscle fibers and two collagenous membranes surrounding them (outer and inner tunic). There is one literature source in which Otwell et al. (1980) presented a sketch of the *Loligo peale* squid mantle with the specifics of all the three layers, as well as the corresponding SEM images [68].

The structural information, especially the unique architecture of collagen fibers in the GSCM, is of special importance in regard to its mechanical properties. The SEM, AFM, and LSM data show that the collagen laminae are arranged in a basketweave manner. We also have studied the structure of the same part from another squid species, *B. magister.* This small squid is easily available as a food product. Its mantle was separated from the muscle layer and studied with AFM. The AFM studies demonstrated a similar structure of the outer tunic for both squid species, despite a significant difference in their sizes. The characteristic features of the GSCM are repeated in the outer tunic of *B. magister* at a smaller scale. It is the structure that was observed in [68] for the *Loligo peale* species. In spite of essential differences in sizes, these squid species have similar morphological and structural features, as well as comparable mechanical characteristics (Table 3).

As the basic component of the squid mantle, collagen is related to the mechanism of the animal’s locomotion. The collagenous membrane of the cephalopod has a basketweave structure that must work as a reinforcing frame in the squid’s body, providing the appropriate strength and stiffness and allowing it to function at high depths.

Indeed, the data of the mechanical tests show rather high values of the tensile strength and Young’s modulus for a biological collagen-based material [41,42,81]. A high value of strain at rupture is also notable. The GSCM mechanical characteristics at the microlevel measured by AFM are also high, which is associated with the tight collagen packing in the material in the form of a basketweave revealed by the microscopical visualization (SEM, LSM, ASM, histological staining). These findings are very important from the viewpoint of the potential GSCM applications in regenerative medicine.

A surgical material must have a good compatibility with the host organism tissues. Our cell experiments with gingival MSC and AlamarBlue, Live/Dead, and PicoGreen assays, as well as the LAL test and preliminary in vivo studies, have demonstrated that the GSCM does not exhibit any cytotoxic properties that testify its good biocompatibility.

The collagenase digestion experiment has additionally confirmed the collagenous nature of the material and proven that it can undergo almost complete destruction in vitro in as soon as 6 h. After the treatment, a non-collagenous amorphous component is left, which binds to picric acid and hematoxylin, but it does not bind to picrosirius red and does not show birefringence. Most likely, this component consists of glycoproteins that bind collagen fibers together, thus providing their corresponding orientation and packing in each layer-lamina and also binding together laminae themselves. However, the presence of this non-collagenous component does not prevent the enzymatic action on collagen fibers in the material that may lead to its biodegradation in vivo.

Thus, the collagen nature, basketweave layered structure, good mechanical properties, absence of cytotoxicity, and ability to biodegrade make the GSCM a prospective candidate for tissue engineering applications.

## 4. Materials and Methods

### 4.1. Material

In this study, we used a commercial material—Aksolagen membrane—provided by the Akses Swiss company (Zug, Switzerland). Aksolagen membrane is a specially treated GSCM of *Dosidicus gigas*. The squid mantle consists of several layers (Figure 15), with the central muscle layer surrounded by two collagenous membranes (outer and inner tunics); the GSCM represents the outer tunic of the mantle.

We also studied the structure of the collagenous membrane of another squid species, a small squid *B. magister.* A frozen squid *B. magister* was purchased in a supermarket, thawed, and the collagenous membrane (outer tunic) was mechanically separated from the muscle layer of the mantle.

The thickness of the GSCM of *D. gigas* measured with a micrometer (a 5–10 N load) was 50 ± 5 μm, and the thickness of the *B. magister* membrane was 25 ± 5 μm.

### 4.2. Amino Acid Analysis

To study the GSCM composition, we conducted the amino acid analysis. The analysis was performed by ion-exchange chromatography, with the post column derivatization, using an L-8800 amino acid analyzer (Hitachi, Ltd., Tokyo, Japan) with a steel Hitachi Ion-Exchange Column 2622SC(PH) (Hitachi, Ltd., Tokyo, Japan) 4.6 × 80 mm. The column temperature was 57 °C, the flow rate was 0.4 mL/min, the charge volume was 50 µL, and the elution regime involved a stepwise gradient of eluents A (AAA PH-1 Buffer—AN0-8706 Merck Hitachi, Tokyo, Japan), B (AAA PH-2 Buffer—AN0-8707 Merck Hitachi, Tokyo, Japan), C (AAA PH-3 Buffer—AN0-8708 Merck Hitachi, Tokyo, Japan), D (AAA PH-4 Buffer—AN0-8709 Merck Hitachi, Tokyo, Japan), and E (0.2 M NaOH solution). As a calibration mixture, standard concentrated amino acid mixtures in ampoules were used (Amino Acid Standard Sigma Aldrich, St. Louis, MI, USA).

To prepare the studied solution, a dry sample was placed in a molybdenum glass ampoule, and 0.3 mL of a freshly prepared hydrolyzing mixture (concentrated hydrochloric and trifluoroacetic acids in a 2:1 ratio with the addition of 0.1% β-mercaptoethanol Sigma Aldrich, St. Louis, MI, USA) was added. The sample was frozen, and the ampoule was evacuated and sealed. The hydrolysis was conducted at 155 °C for 1 h. After the hydrolysis, the ampoule was cooled, opened, and the content was quantitatively transferred (0.1 mL of water twice) in a plastic 1.5 mL tube, then the hydrolyzing mixture was completely removed with a CentriVap vacuum concentrator (Labconco corporation, Kansas, MO, USA) at 50 °C. The residual acids were removed by repeating twice the procedure of evaporation of small water portions (0.1 mL) added to the dry residue at 50 °C. Then, 0.1 N HCl was added to the dry residue, the mixture was centrifugated, and 0.1 N HCl was added to the supernatant in a 10:1 ratio.

### 4.3. Collagen Molecular Weight Estimation (SDS-PAGE)

Following the collagen extraction, the protein concentration was evaluated by the gravimetric analysis. The sample was ≈100-fold concentrated by ultrafiltration on a Microcon Centrifugal filter unit with a 10 kDa molecular cut-off (MRCPRT010, Millipore, Burlington, MA, USA) to obtain the final collagen at 10 mg/mL. Collagen from GSCM and Type I collagen from the cattle dermis were isolated using a protocol described in [82], while Type II collagen was isolated from the tracheal cartilage by a protocol described in [83] omitting the use of pepsin. An amount of 10 µg of the proteins were diluted with an SDS-loading buffer supplemented with 100 mM DTT (20710, SERVA, Heidelberg, Germany) and heated at 95 °C for 5 min. The samples were resolved by denaturing polyacrylamide gel electrophoresis in 5% stacking and 8% separating gel using a Mini-PROTEAN Electrophoresis System (Bio-Rad, Hercules, CA, USA). The electrophoresis running conditions were as follows: at 15 mA, until samples reached the separating gel, then at 30 mA until the front reached 0.5 cm from the gel edge. The gel was stained with Coomassie Blue R-250 (35051, SERVA, Heidelberg, Germany) and scanned with a ChemiDoc Imaging System (Bio-Rad, Hercules, CA, USA). The molecular weights of the visual bands were calculated in the ImageLab software against the molecular weight standards (Spectra Multicolor High Range Protein Ladder, SM1851, Fermentas, Waltham, MA, USA).

### 4.4. IR-Spectroscopy

The FTIR analysis of the initial components was carried out using a Spectrum Two FT-IR Spectrometer (PerkinElmer, Waltham, MA, USA) in the Attenuated Total Reflectance (ATR) mode. The spectrometer features were as follows: a high-performance, room-temperature LiTaO3 MIR detector, a standard optical system with KBr windows for the data collection over a spectral range of 8300–350 cm^−1^ at a resolution of 0.5 cm^−1^. All the spectra were initially collected in the ATR mode and converted into the IR transmittance mode. The spectra of collagens were normalized using the intensity of the Amid I band as the internal standard.

### 4.5. Differential Scanning Calorimetry (DSC)

Differential scanning calorimetry (DSC) measurements were performed using an STA 6000 simultaneous thermal analyzer (PerkinElmer, Waltham, MA, USA). Samples for DSC experiments (about 10 mg) were encapsulated in standard PerkinElmer pans and heated in a nitrogen medium at a gas flow rate of 20 mL/min and a linear heating rate of 10 °C/min.

### 4.6. Shrinkage Temperature

A sample with the sizes of 3 × 15 mm was placed in a special calibrated glass tube and immersed in a vessel filled with distilled water. The vessel was placed in a water bath. The water bath was heated from room temperature to the moment of the sample shrinkage (~60 °C). The shrinkage temperature was determined as the temperature at which the beginning of the sample shrinkage was detected. The experiment was repeated thrice.

### 4.7. Histological Study

Intact and collagenase-treated fragments of the GSCM of *D. gigas* were fixed in a 10% solution of neutral formalin, and 4 µm-thick histological sections were prepared using a standard technique.

The prepared sections were stained with hematoxylin and eosin and picrosirius red to reveal the collagen composition. The prepared slides were studied by optical (bright-field, phase contrast and polarized light) microscopy, and the images were captured with a LEICA DM4000 B LED microscope equipped with a LEICA DFC7000 T digital camera, using the LAS V4.8 software (Leica Microsystems, Heerbrugg, Switserland).

### 4.8. Scanning Electron Microscopy (SEM)

The GSCM structure was visualized using an EVO LS10 scanning electron microscope (Carl Zeiss Microscopy GmbH Jena, Germany). Two techniques for sample preparation and visualization were used.

The first protocol allowed general images of the samples in the most native state. Naturally dried samples were attached to the microscope stage with a special carbon adhesive tape. The observations were conducted in the low vacuum regime (EP, 70 Pa) at the accelerating voltage of 20 kV and the current of 94 pA per sample. A detector for back-scattered electrons (BSE) was used. The images were obtained with the resolutions of 473.1 nm/px and 508.8 nm/px. To achieve a satisfactory resolution during back-scattered electrons observations, a working distance of 4.5 mm was used.

For detailed evaluation of the structure, samples were fixed in neutral glutaric aldehyde, dehydrated (battery of alcohols from 20% to 97% and acetone), dried bypassing the critical point of CO_2,_, and coated with an Au-Pd alloy. The so-prepared samples were attached to the microscope stage providing the charge outflow from the coated surface. The observations were conducted in the high vacuum regime at the accelerating voltage of 21 kV and the sample current of 19 pA. The microtopography images were obtained using the detector for secondary electrons (SE). The 3072 × 2304 px images were captured with the resolutions of 89.89 nm/px and 2697 nm/px.

### 4.9. Laser Scanning Microscopy (Second Harmonics Generation, SHG Signal)

The study was performed using a LSM 880 NLO laser scanning microscope (Carl Zeiss Microscopy GmbH Jena, Germany) equipped with a tunable Ti:Sa MaiTai HP laser (Spectra-Physics, Milpitas, CA, USA) with a pulse duration of less than 100 fs. The wavelength of 800 nm was used for the study, and the registration of the SHG signal was performed in the range of 370–420 nm. The power of the probing radiation was about 9 mW. The images were obtained using an oil immersion objective with the 40× magnification and numerical aperture of 1.3. The field of view was 212 × 212 µm, and the resolution of images was 1024 × 1024 pixels. A series of images (z-stack) was acquired from the sample surface into the depth with the step of 9 µm, the orientation being parallel to the surface. For the convenience of perception, the acquired images were presented in the green palette.

### 4.10. Atomic Force Microscopy (AFM)

The morphological AFM studies of the surface were performed using an atomic force microscope (BioScope Resolve, Bruker, Billerica, MA, USA) combined with an Axio Observer inverted optical microscope (Carl Zeiss Microscopy GmbH Jena, Germany). A ScanAsyst Air cantilever (Bruker, Billerica, MA, USA) was used with a nominal spring constant k = 0.4 N/m and a nominal tip radius r = 2 nm, and scanning was performed on air in the PeakForce QNM regime. The collagen structures’ periodicity was estimated with the Section function of the NanoScope Analysis v1.9 software (Bruker, Billerica, MA, USA).

### 4.11. Uniaxial Stretching Test

The uniaxial stretching tests for dry and hydrated samples were conducted using a Mach-1 v500c mechanical tester (Biomomentum, Laval, QC, Canada). For the hydration, samples were immersed in distilled water for 20 min. The measurements were also performed in distilled water. Dumbbell-shaped fragments of the dry and hydrated GSCM were cut both in the tangential and radial directions in respect to the whole material area (with the circle diameter of 30 cm). The working area of the fragments had the length of 15 mm and width of 5 mm. The dry material thickness was 45 µm, while the thickness of the hydrated material was 60 µm. Before the test, the mechanical tester was calibrated using a standard sample provided by the manufacturer. Both ends of the experimental sample were tightly gripped in the clamps followed by gradual elongation at room temperature (25 °C) at a constant rate of 0.1 mm/s until rupture. The mechanical parameters were calculated from the stress-strain curves according to the manufacturer’s protocol. The data were averaged over 3 or more tests.

### 4.12. Micromechanics by AFM

The mechanical properties of the samples’ surface were studied in fluid (distilled water) at room temperature (25 °C), after 20 min of hydration, using an atomic force microscope (BioScope Resolve, Bruker, Billerica, MA, USA). The sample micromechanics was obtained in the regime of nanoindentation over a preset map of 50 × 50 µm with the 32 × 32 pixels resolution, as described in [84]. A ScanAsyst Fluid cantilever (Bruker, Billerica, MA, USA) with a nominal spring constant of 0.95 N/m and a nominal tip radius of 50 nm was precalibrated using a standard titanium sample. The deflection sensitivity was calibrated in the same conditions using a sapphire standard sample. The data were processed using the NanoScope Analysis v1.9 software(Bruker, Billerica, MA, USA) and averaged over 12 measurements.

### 4.13. In Vitro Cytotoxicity Assays

The biocompatibility and cytotoxicity tests were performed using the primary culture of mesenchymal stromal cells (MSCs) isolated from human gingival mucosa as described in [85]. The cells were cultivated in the medium that contained Dulbecco’s Modified Eagle’s Medium (DMEM)/F12 (1:1, Biolot, St. Petersburg, Russia), 10% fetal calf serum (HyClone, Logan, UT, USA), L-glutamine (5 mg/mL, Gibco, Gaithersburg, MD, USA), insulin–transferrin–sodium selenite (1:100, Biolot, St. Petersburg, Russia), bFGF (20 ng/mL, ProSpec, Rehovot, Israel), and gentamycin (50 μg/mL, Paneco, Moscow, Russia). Isolated cells were routinely checked with a SH800S microfluidic flow cytometer (Sony Biotechnology, San Jose, CA, USA) for the presence of mesenchymal surface markers (CD90, CD73, CD105) and absence of hematopoietic and endothelial markers (CD45, CD34, CD11b, CD19 and HLA-DR), according to [86]. The cells were cultivated in the standard conditions of 37 °C and 5% CO_2_.

The cytotoxicity was analyzed via the elution and contact tests. In the first case, the extracts of the GSCM were prepared according to recommendations of ISO 10993-12. Briefly, 5000 cells per well of a 96-well plate were seeded 24 h before adding the extracts. To prepare extracts, GSCM films were incubated in the culture medium for 24 h at 37 °C. The thickness of a film was less than 0.5 mm, and, therefore, in accordance with ISO 10993-12, the required sample’s area was to be treated in a volume of 1 mL is 6 cm^2^. Cells were exposed to the maximum concentration of the extract (6 cm^2^/mL) and its serial twofold dilutions. We used serial two-fold dilutions of 1.5 mg/mL sodium dodecyl sulphate (SDS) in a standard culture medium as a positive control. Cells cultivated in the standard culture medium were applied as a negative control. After 24 h of cultivation with the extract, SDS, or culture medium, the cell viability was assessed either with the AlamarBlue cell viability reagent (Invitrogen, Waltham, MA, USA) or with the Quant-iT PicoGreen kit (Invitrogen, Waltham, MA, USA). For the AlamarBlue metabolic activity assay, the cell culture medium was replaced with a 10% reagent solution and incubated for 2 h. Then, the fluorescence of samples was measured using a Victor Nivo spectrofluorometer (PerkinElmer, Waltham, MA, USA) at a 530 nm excitation wavelength and a 590 nm emission wavelength. The DNA amount was evaluated with the PicoGreen assay after 3 freeze-thaw cycles aimed at releasing DNA, following the manufacturer’s instructions. The samples’ fluorescence was estimated with the spectrofluorometer at a 480-nm excitation wavelength and a 520-nm emission wavelength.

For the contact cytotoxicity, 20,000 cells were seeded on a surface of the 1 cm^2^ GSCM films and cultivated for 3 days. Cells seeded on the culture plastic (monolayer culture) served as a control. Afterwards, the metabolic activity and DNA amount were measured as described above.

The morphology and viability of the cells seeded on the GSCM was visualized with the Live/Dead assay. Briefly, live cells were stained with calcein-AM (Sigma-Aldrich, St. Lois, MO, USA), dead cells were stained with propidium iodide (Thermofisher, Waltham, MA, USA), and nuclei were stained with Hoechst 33,258 (Thermofisher, Waltham, MA, USA). The images were obtained by laser confocal scanning microscopy using a LSM 880 instrument with Airyscan (Carl Zeiss Microscopy GmbH Jena, Germany).

All the samples were triplicated (plate wells for extract cytotoxicity and film samples for the contact cytotoxicity and Live/Dead assay).

### 4.14. Resistance to Collagenase

The susceptibility to proteolytic degradation was studied in a Collagenase A (from C histolyticum) solution. Approximately 4 mg (dry weight in triplicates) of the sample were weighed. To the weighed samples, 0.5 mL aliquots of a 2.5 mg/mL Collagenase A solution in the Tris buffer (50 mmol/L, pH 7.5) containing 10 mmol/L calcium chloride and 0.02 mg/mL sodium azide (Paneco, Moscow, Russia) were added. The samples were incubated at 37 °C for 6 h. Then, the samples were centrifuged at 605 g (3000 RPM) for 90 s (a MiniSpin microcentrifuge by Eppendorf Corporation, Hamburg, Germany). We used a low rotation speed and a short time of centrifugation in order to better preserve the structure integrity for the following histological analysis. Then, the material was washed from the residual collagenase with distilled water. The precipitate was carefully transferred using a micropipette to a coverslip for the following drying in an oven at 50 °C for 20 h. Then, the dry residue was weighed using a WXTE ultramicrobalance (Mettler Toledo GmbH Urdorf, Switzerland). Finally, the weight loss was calculated by a paired comparison before and after the treatment.

### 4.15. LAL Test

The GSCM film was cut into 5*5-mm pieces under aseptic conditions. The extracts were prepared in 1 mL of endotoxin-free water by continuous shaking for 24 h at 50 °C. The endotoxin concentrations were measured using the Chromogenic Endotoxin Quantitation Kit (Thermo Fisher Scientific, Waltham, MA, USA) in accordance with the manufacturer’s instruction. Briefly, we mixed 50 μL of the extract or the endotoxin standard dilution (0.1, 0.25, 0.5, 0.1 U/mL) and 50 μL of endotoxin-specific Limulus Amebocyte Lysate (LAL) reagent in a well of a 96-well plate. The mixture was incubated for 10 min at 37 °C and then 100 μL of the chromogenic substrate was added and incubated for 6 min at 37 °C. The reaction was inhibited by adding 100 μL of 25% acetic acid. The absorbance was measured at a wavelength of 405 nm using a microplate Victor Nivo spectrofluorometer (PerkinElmer, Waltham, MA, USA). The minimal detection level of the kit used was 0.1 EU/mL (EU—unit of measurement for endotoxin activity).

## 5. Conclusions

The literature analysis shows that the GSCM material has been very poorly studied, and no application in tissue engineering has been discussed so far. The results of our studies on the GSCM composition, structure, mechanical characteristics, cytotoxicity, and biodegradability testify that the GSCM of *D. gigas* is characterized by a high tensile strength and elasticity, along with a peculiar basketweave collagen structure and biocompatibility that allows the assumption that this material may be applicable in a number of tissue engineering fields (e.g., wound care materials, scaffolds for restoration of the musculoskeletal system, repair of hernias and the prolapse of pelvic organs, dental membranes, and other applications requiring good mechanical properties and slow degradation of the implanted material).

Upon the comparison with other squid species (in particular, *B.magister)*, one may conclude that the GSCM structure is represented by a typical reinforcing mesh consisting of collagen structures and providing the high strength and Young’s modulus. However, since the Giant squid *D. gigas* has a large size of the mantle and, respectively, a large lateral size of the GSCM, this material is more advantageous from the processing viewpoint.

## Figures and Tables

**Figure 1 marinedrugs-19-00679-f001:**
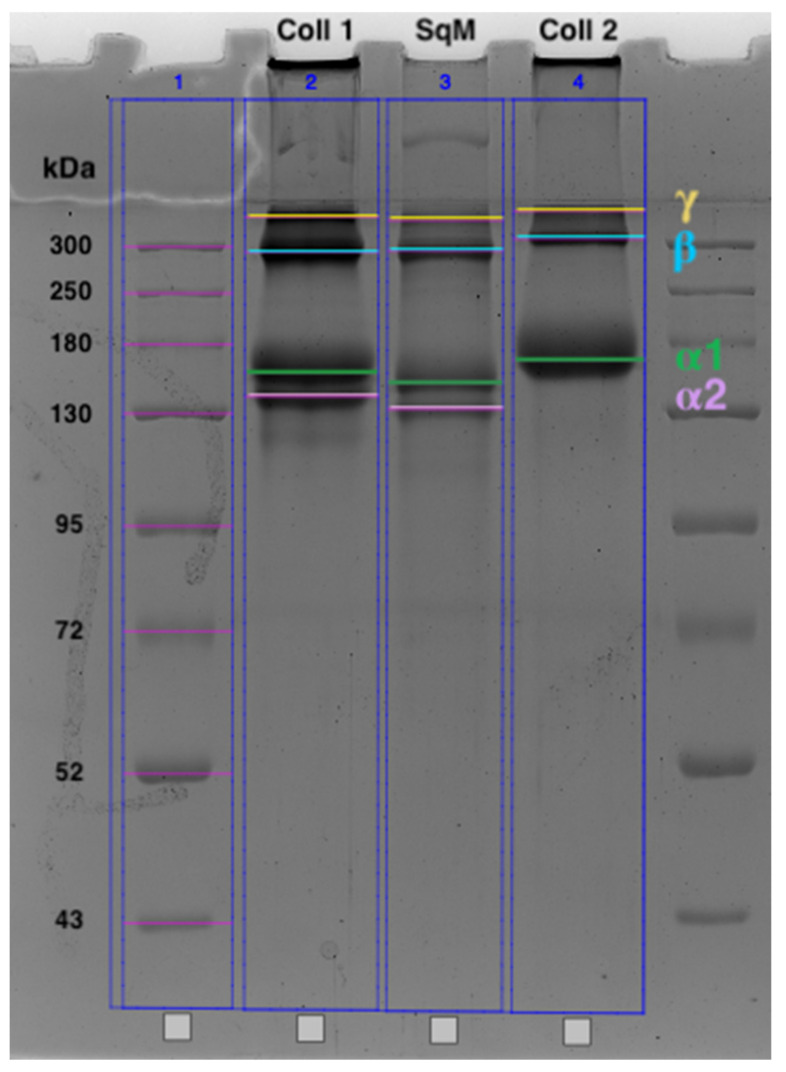
Evaluation of the GSCM collagen chains’ electrophoretic mobility in 8% PAAG under the denaturing and reducing conditions. Collagen Type I (Coll 1) and Type II (Coll 2) were used as collagen standards. SqM—collagen extracted from the GSCM. The high range protein ladder bands are shown in kDa.

**Figure 2 marinedrugs-19-00679-f002:**
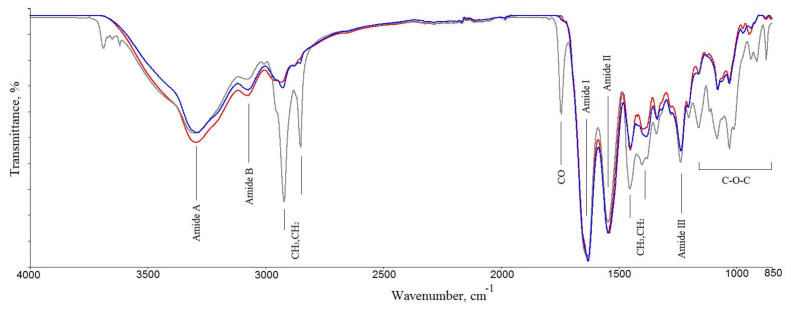
FT-IR spectra of the GSCM is the blue curve; collagen Type I is the gray curve; collagen Type II is the red curve.

**Figure 3 marinedrugs-19-00679-f003:**
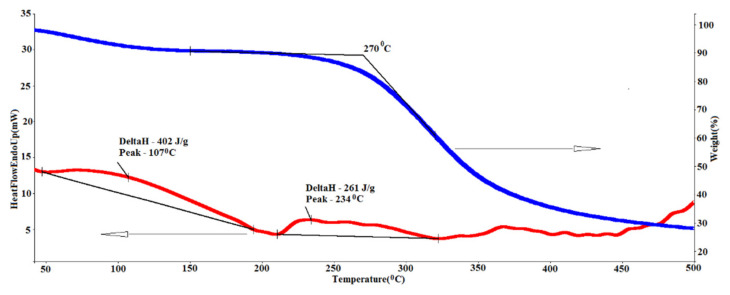
TG (blue) and DSC (red) curves for the GSCM.

**Figure 4 marinedrugs-19-00679-f004:**
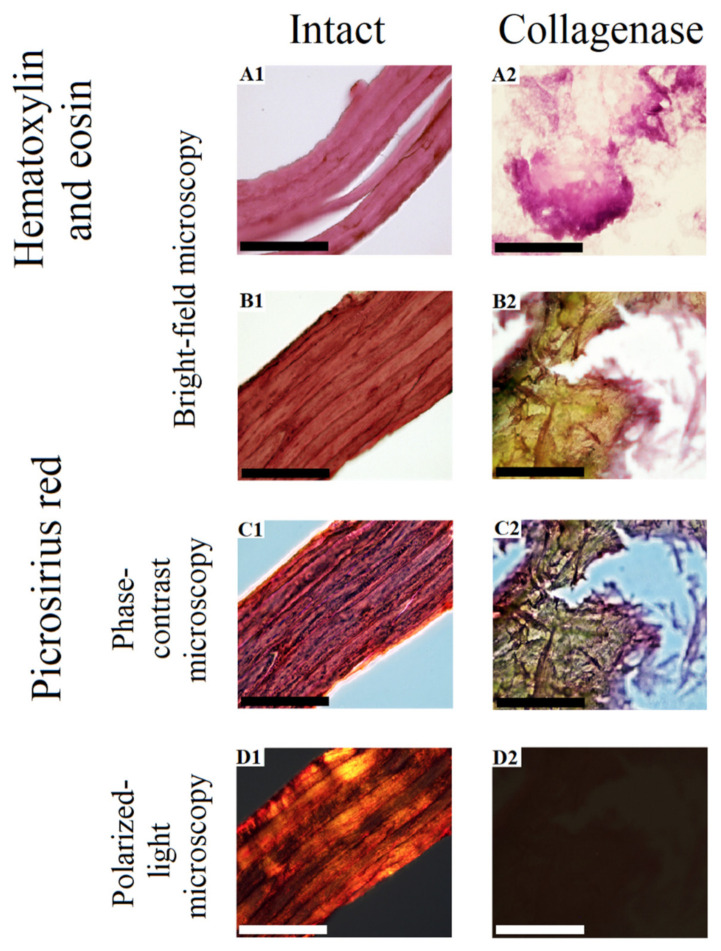
Morphological and optical characteristics of the GSCM before and after the collagenase treatment. (**A1**) As seen at a cross-section, the GSCM consists of parallel uniform pink (eosinophilic) layers—“laminae”; (**A2**) lysis of the material with homogenization, loss of crisp contours, and appearing purple (basophilic) regions; (**B1**) predominantly red staining of laminae with regions of poorly visible fine-fibred structure; (**B2**) loss of the red and appearance of yellow (picrinophilic) staining in most parts of the material with single loose and multidirectional red collagen fibers; (**C1**) a somewhat more visible fibrillar structure of the material than that in (**B1**); (**C2**) scattered collagen fibers among the picrinophilic material are more visible than they are in (**B2**); (**D1**) laminae produce a bright yellow-green, yellow-orange, and orange-red glow due to the collagen fibers within their structure; (**D2**) no material glow was noted; ×1000 (Scale bar = 50 µm).

**Figure 5 marinedrugs-19-00679-f005:**
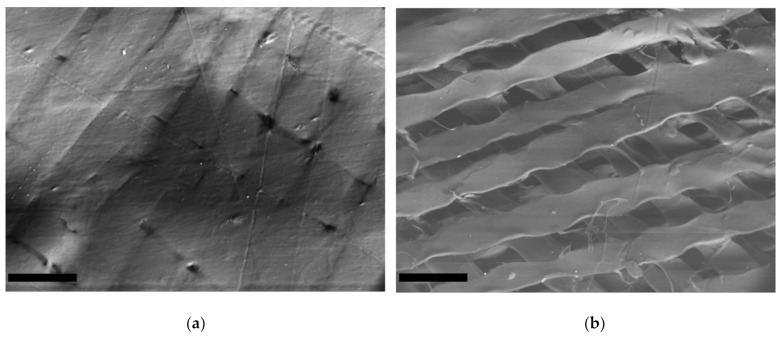
SEM-BSE images of the GSCM surface. (**a**) Native surface, and (**b**) a region inside a fracture zone of the material (Scale bar = 100 µm).

**Figure 6 marinedrugs-19-00679-f006:**
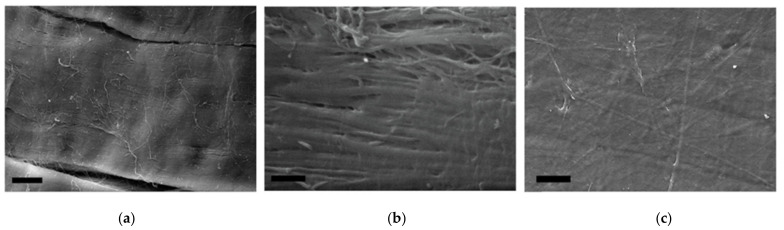
Microtopography of the dried GSCM (SEM-SE). (**a**) the surface of a lamina comprising the reinforcing layer (Scale bar = 10 µm), (**b**) the enlarged fragment of the lamina surface (Scale bar = 1 µm), and (**c**) the layer covering the GSCM reinforcing layers (Scale bar = 20 µm).

**Figure 7 marinedrugs-19-00679-f007:**
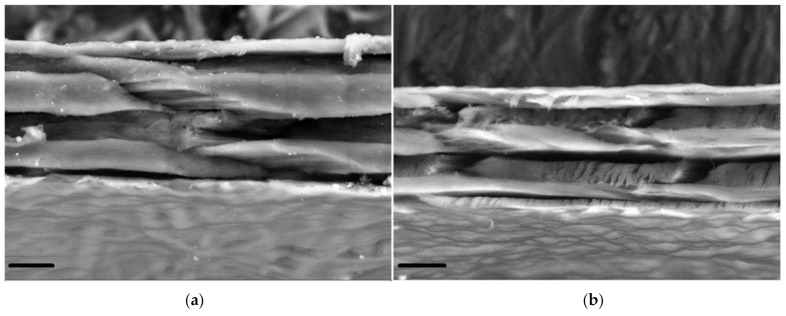
A SEM-BSE image of the GSCM cross-section. (**a**,**b**) Two different regions (Scale bar = 10 µm).

**Figure 8 marinedrugs-19-00679-f008:**
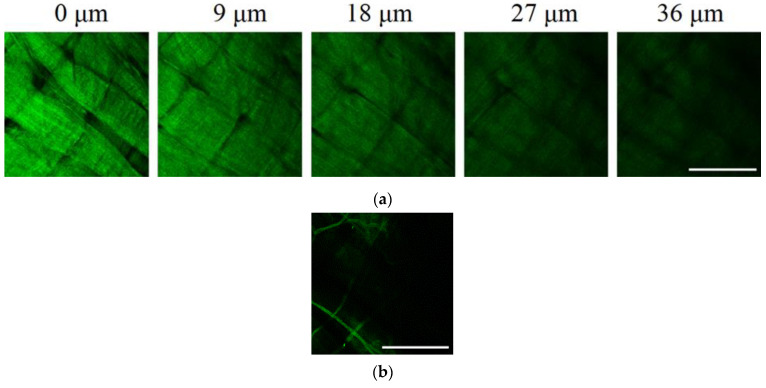
SHG images of the GSCM. (**a**) Collagen bundles in the form of laminae at different depths; (**b**) collagen in the form of cords at the sample’s surface (Scale bar = 100 µm). The SHG signal from collagen is marked by green.

**Figure 9 marinedrugs-19-00679-f009:**
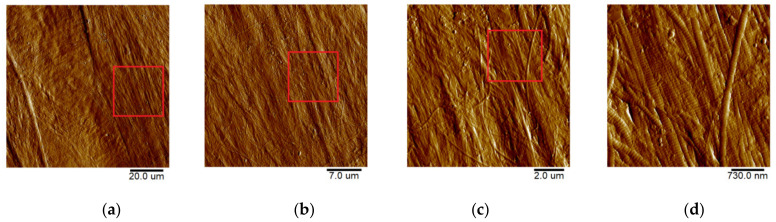
AFM topography of the GSCM with the sequential decrease of the scan size from the left to right image (increase of resolution): (**a**) 100 × 100 µm; (**b**) 30 × 30 µm; (**c**) 10 × 10 µm; (**d**) 3 × 3 µm. The samples’ topography is presented using the Peak Force Error for the better detail resolution.

**Figure 10 marinedrugs-19-00679-f010:**
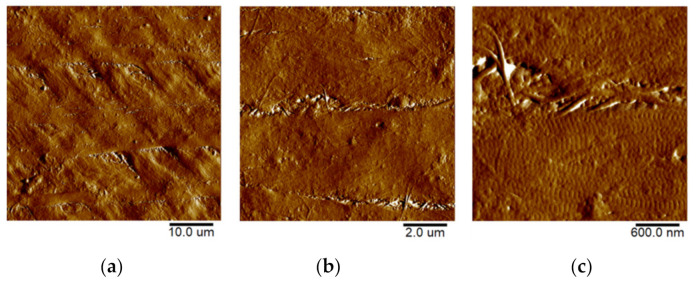
AFM topography of the collagenous membrane of a *B. magister* squid. From the left to right image (increase of resolution): (**a**) 50 × 50 µm; (**b**) 10 × 10 µm; (**c**) 3 × 3 µm. The samples’ topography is presented using the Peak Force Error for the better detail resolution.

**Figure 12 marinedrugs-19-00679-f012:**
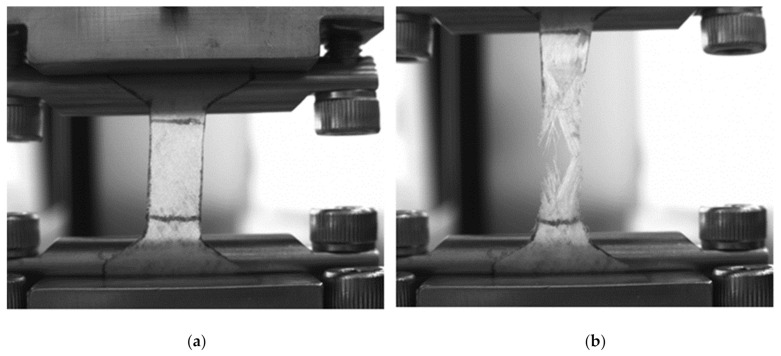
A uniaxial stretching test: (**a**)—start, (**b**)—end of test.

**Figure 13 marinedrugs-19-00679-f013:**
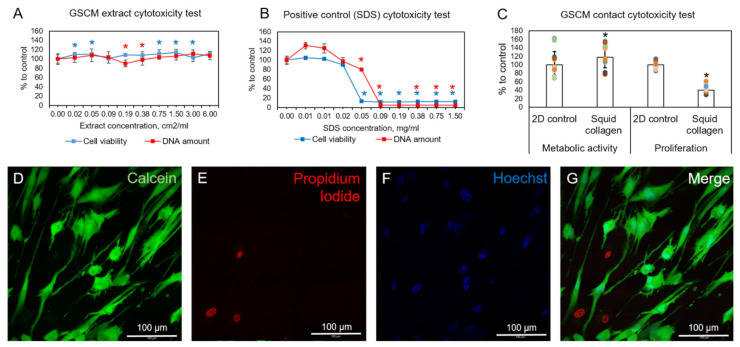
(**A**)—Elution test: AlamarBlue cytotoxicity assay and PicoGreen DNA assay of the GSCM extracts, 3 days of MSCs’ cultivation, 5000 cells per well. (**B**)—AlamarBlue cytotoxicity assay and PicoGreen DNA assay of SDS (positive control), 3 days of MSCs cultivation, 5000 cells per well. (**C**)—AlamarBlue contact cytotoxicity assay and PicoGreen DNA assay of cells adhered to culture plastic (2D control) or GSCM, 3 days of cell cultivation, 20,000 cells per well. * *p* < 0.05 relative to other groups. (**D**–**G**)—Live/Dead cell viability assay with nuclei staining (Hoechst, blue); live cells are stained with Calcein AM (green), and dead are stained with propidium iodide (red). At 7 days of cultivation, laser scanning confocal microscopy, scale bar is 100 μm.

**Figure 14 marinedrugs-19-00679-f014:**
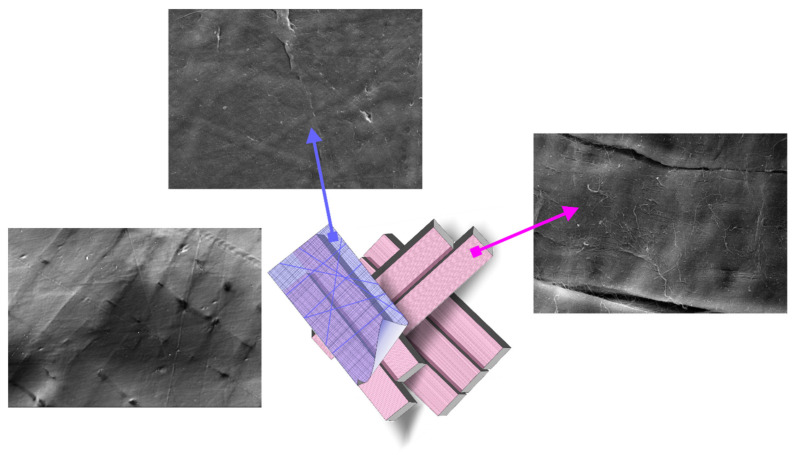
A possible concept of the arrangement of collagen fibers in the GSCM based on the SEM findings.

**Figure 15 marinedrugs-19-00679-f015:**
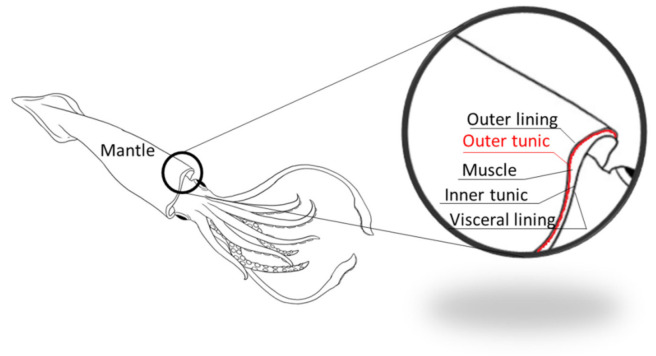
The *D. gigas* mantle and its inner structure.

**Table 1 marinedrugs-19-00679-t001:** Mechanical properties of some collagen-based tissues and materials for tissue engineered constructs.

Tissue/Material	Treatment	Tensile Test Data		References
		Ultimate Tensile Strength, UTS (MPa)	Young’s Modulus (MPa)	
Human Skin		27.2 ± 9.3 MPa	98.9 ± 97 MPa	[23,29]
Reconstructed anterior cruciate ligament (ACL) rabbit model	Glutaraldehyde cross-linked prostheses	26 MPa		[30]
Reconstructed anterior cruciate ligament (ACL) rabbit model	Carbodiimide cross-linked prostheses	12 MPa		[30]
Reconstructed anterior cruciate ligament (ACL) rabbit model	Sham-operated controls	49 ± 20 MPa		[30]
Human patellar tendon		60–100 MPa	300–400 MPa	[31]
Human native rotator cuff tendon		11.5 ± 5 MPa	50–170 MPa	[32]
TSPC seeded knitted silk–collagen sponge scaffold for functional shoulder repair rabbit model	TSPC seeded	Control group 5.9 ± 1 MPa; TCPC group 8.3 ± 1.5 MPa	Control group 44.3 ± 12.1 MPa; TCPC group 67.8 ± 14.6 MPa	[33]
Human Achilles tendon		40 ± 8 MPa	1600 ± 200 MPa	[34]
Rabbit Achilles tendon		4.5 MPa	45 MPa	[14]
Human fibrocartilage		10 MPa		[26]
Human compact bone		0.03 MPa	15,000 MPa (Depending on type and size of the bones)	[35]
Human vaginal tissue		0.82–2.62 MPa		[36]
Human cornea		3.81 ± 0.4 MPa		[37]
DBP, decellularized bovine pericardium		Along 23 MPa Across 20 MPa	Along 120 MPa Across 50 MPa	[38]
Normal human skin (NHS)		2.8 MPa		[39]
ASC from bovine hide scaffolds by electrospinning		0.4 MPa		[39]
Un-crosslinked collagen film from bovine tendon		10 ± 0.5 MPa		[40]
Un-crosslinked collagen film from (Coll type I)		37.7 ± 4.5 MPa	1100 ± 100 MPa	[41]
Collagen films from rat tail (Coll type I)		100 MPa	27 MPa	[42]
Chitosan-AS collagen biofilms from mantle *D. gigas*		33.5 ± 4 MPa		[43]
Collagen fiber films from cattle skin		Dry 17.25 ± 0.07 MPa Wet 2.61 ± 0.05 MPa		[44]
Fresh (non-treated) pulmonary heart valves pigs		0.5 ± 0.2 MPa		[45]

**Table 2 marinedrugs-19-00679-t002:** Amino acid composition of the *D. gigas* CM.

Name of Amino Acids	Abbreviation	Letter Code	Molecular Mass, g/mol	Residues per 1000 Residues	w% *
Alanine	Ala	A	89.094	86.2	6.87
Arginine	Arg	R	174.203	56.4	8.79
Aspartic acid	Asp	D	133.104	62.9	7.49
Cysteine	Cys	C	121.154	3.5	0.74
Glutamic acid	Glu	E	147.131	86.4	11.38
Glycine	Gly	G	75.067	330.0	22.18
Histidine	His	H	155.156	7.7	1.07
Hydroxyproline	Hyp	O	131.131	86.3	10.13
Hydroxylysine	Hyl		162.187	10.3	1.5
Isoleucine	Ile	I	131.175	13.9	1.64
Leucine	Leu	L	131.175	29.5	3.47
Lysine	Lys	K	146.189	14.0	1.83
Methionine	Met	M	149.208	10.4	1.39
Phenylalanine	Phe	F	165.192	11.1	1.64
Proline	Pro	P	115.132	91.3	9.4
Serine	Ser	S	105.093	41.1	3.86
Threonine	Thr	T	119.119	27.9	2.97
Tyrosine	Tyr	Y	181.191	6.4	1.04
Valine	Val	V	117.148	24.9	2.61
Total				1000	
	Hyp/Hyl			8.4	

* Percentage of amino acids to the mass of the test sample.

## Data Availability

All the data supporting the conclusions of this article are included in this article.

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
