# Peer review of "A Collagen Basketweave from the Giant Squid Mantle as a Robust Scaffold for Tissue Engineering"

_marinedrugs, 2021, doi:10.3390/md19120679_

Round 1

Reviewer 1 Report

The criticisms have been addressed properly by the authors.

Reviewer 2 Report

The revised manuscript is OK for acceptance now

This manuscript is a resubmission of an earlier submission. The following is a list of the peer review reports and author responses from that submission.

Round 1

Reviewer 1 Report

The manuscript from Frolova et al. is a study investigating the chemico-physical and biochemical characteristics of a commercial giant squid-derived collagen membrane, distributed by a Swiss Company for cosmetic use (Akses Swiss Gmbh). The authors indeed demonstrate the collagenous nature of the membrane obtained from this marine invertebrate by various and updated investigating methods. Although, being already on the market and sold as a “marine collagen biomaterial” it does not seem a big novelty. I would have expected a more detailed investigation on the biomedical properties which are indeed just outlined by the authors in this paper. Biocompatibility and cytotoxicity are investigated only on one cell type, Mesenchymal Stem Cells, but since the membrane is suggested to be useful for various biomedical problems, namely vascular, skin, bone, cartilage and so on, it would have been interesting to see a biocompatibility study on these types of tissues using the relative cell types to assess in which conditions it would give the best performance. No details on the membrane preparation are given, and no study on the bacterial adhesion and contamination have been performed so it’s not possible to comment on the biosafety of use in biomedical applications.

For what concerns the results on the viability test on MSCs a lot of information is lacking:

In the results section which types of cells were used is missing, how many were seeded on the membranes, to assess cytotoxicity which concentration of the extract was used (cm2/ml does not say anything, if not explained). How the membranes were obtained? Cross-linking agents were used? It has been tested if the cross-linking agents may be released from the membranes, during long time soaking, to cause cytotoxicity?

Adhesion and proliferation are two different aspects to investigate: cell adhesion should be measured at a shorter time point (less than 24 hrs), while cell proliferation at longer time points (3-6 days) to assess if cells after adhesion are able to proliferate on the matrix.

Since the authors used MSCs, why did they not investigate the osteogenic and chondrogenic potential of the collagenous membranes? It would seem to fit with the biomedical use proposed by the authors.

Are the membranes safe from bacterial contamination and are bacteria able to adhere and proliferate on the membranes? This should be investigated to assess the biosafety of the material for tissue engineering.

Are the images of Fig. 13 from fluorescence or confocal microscopy? The legend of this figure is not well explained.

I also noticed the following minor points, but the list could be longer:

In the Introduction there is a gap in the reference numbers going from ref 25 directly to ref 43. Please check the order of your references.

When there is a citation from another paper please name the first author and then the reference number in brackets.

In the legend of Fig. 3 you should indicate which colour of the curve corresponds to which measurement.

In the results section 2.2.3 Shrinkage temperature, talking about collagen hydration measurement, is it similar to human collagen in this regard, or it is different?

In the Materials and methods how the collagen was extracted from the matrix to perform SDS-PAGE is not explained, in the cytotoxicity section information is lacking, how many cells were used for each experiments, how many replicates etc.

Author Response

Dear Reviewer,

Thank you for your decision to allow the manuscript revision to meet the Reviewers’ requests. In accordance with the Reviewers’ recommendations, we have performed a revision of the manuscript.

Below, we present the Reviewers’ comments in Italics and our response to them in plain text. All the revisions made in the text of the manuscript are seen green color type submitted as a Review Only Supporting Information.

The manuscript from Frolova et al. is a study investigating the chemico-physical and biochemical characteristics of a commercial giant squid-derived collagen membrane, distributed by a Swiss Company for cosmetic use (Akses Swiss Gmbh). The authors indeed demonstrate the collagenous nature of the membrane obtained from this marine invertebrate by various and updated investigating methods. Although, being already on the market and sold as a “marine collagen biomaterial” it does not seem a big novelty. I would have expected a more detailed investigation on the biomedical properties which are indeed just outlined by the authors in this paper. Biocompatibility and cytotoxicity are investigated only on one cell type, Mesenchymal Stem Cells, but since the membrane is suggested to be useful for various biomedical problems, namely vascular, skin, bone, cartilage and so on, it would have been interesting to see a biocompatibility study on these types of tissues using the relative cell types to assess in which conditions it would give the best performance. No details on the membrane preparation are given, and no study on the bacterial adhesion and contamination have been performed so it’s not possible to comment on the biosafety of use in biomedical applications.

We thank the Reviewer for the comprehensive commentaries. Collagen membranes have been shown to be an excellent candidate for tissue engineering, since they are highly biocompatible and provide crucial cues for cells. Testing various cell types may be interesting in engineering particular tissues. However, this study aimed, first-of-all, to provide the general characteristics of GSCM and demonstrate that this material can be of interest for tissue engineering applications. The MSC primary culture was chosen because MSCs are commonly applied in tissue engineering (10.3390/cells8080886, 10.1038/s41598-017-17286-1, 10.1117/1.JBO.25.4.048001) and were shown to be more sensitive to toxic agents than 3T3 or L929 cell line (10.3390/jfb11030058, 10.1016/j.actbio.2020.08.012, 10.1002/jbm.a.36054). Further experiments with other cell types are planned to be performed and presented in the next manuscript.

For what concerns the results on the viability test on MSCs a lot of information is lacking:

In the results section which types of cells were used is missing, how many were seeded on the membranes, to assess cytotoxicity which concentration of the extract was used (cm2/ml does not say anything, if not explained).

We added the required information to the Materials and Methods and Results sections and expanded the description for Fig. 13.

How the membranes were obtained? Cross-linking agents were used? It has been tested if the cross-linking agents may be released from the membranes, during long time soaking, to cause cytotoxicity?

Adhesion and proliferation are two different aspects to investigate: cell adhesion should be measured at a shorter time point (less than 24 hrs), while cell proliferation at longer time points (3-6 days) to assess if cells after adhesion are able to proliferate on the matrix.

To avoid misunderstanding, we made the required changes and used the term «contact cytotoxicity». Therefore, we provided the information on metabolic activity (AlamarBlue) and proliferation activity (PicoGreen) of the cells cultivated on the GSCM surface for 3 days.

Since the authors used MSCs, why did they not investigate the osteogenic and chondrogenic potential of the collagenous membranes? It would seem to fit with the biomedical use proposed by the authors.

The aim of this paper was to provide the basic GSCM characteristics required to consider it as a possible biomaterial in tissue engineering. Using the MSC culture, we showed that the studied material is non-cytotoxic. So, surely, further experiments will be performed to expand its future application in engineering particular tissues and organs.

Are the membranes safe from bacterial contamination and are bacteria able to adhere and proliferate on the membranes? This should be investigated to assess the biosafety of the material for tissue engineering.

Are the images of Fig. 13 from fluorescence or confocal microscopy? The legend of this figure is not well explained.

These images were obtained using the laser scanning confocal microscopy. We have expanded the description of Fig. 13 with the required details.

I also noticed the following minor points, but the list could be longer:

In the Introduction there is a gap in the reference numbers going from ref 25 directly to ref 43. Please check the order of your references.

In fact, there is no gap in the reference numbers, since refs. 26-42 are in Table 1 of the Introduction section.

When there is a citation from another paper please name the first author and then the reference number in brackets.

We corrected all such citations as suggested by the Reviewer.

In the legend of Fig. 3 you should indicate which colour of the curve corresponds to which measurement.

We added the corresponding description to the caption of Figure 3.

In the results section 2.2.3 Shrinkage temperature, talking about collagen hydration measurement, is it similar to human collagen in this regard, or it is different?

In principle, we see no reason for human collagen to behave differently. It would be difficult to arrange the corresponding experiment since the access to human collagen is restricted due to ethical reasons.

In the Materials and methods how the collagen was extracted from the matrix to perform SDS-PAGE is not explained,

We added the information concerning the collagen extraction to the Materials and Methods section (please see 4.3 on page 18).

in the cytotoxicity section information is lacking, how many cells were used for each experiments, how many replicates etc.

As mentioned above, we added the information concerning the cell type and concentration, extract concentration to the Materials and Methods and Results sections, and also supplemented the description of Fig. 13 with some experiment details. We also added a broader explanation of the extract concentrations used (please see 4.13 on page 21).

Reviewer 2 Report

This study topic is interesting and important. Overall, this report has good quality and the authors have provided some results to support the significance of this study. Reasonable revisions are needed before acceptance.

Comments and suggestions:

1, the reason for choosing/designing this study need to be explained more

2, an illustration figure about this study and giant squid mantle is suggested

3, degradation test is suggested if possible

4, more background and refs about the medical applications of collagen scaffolds are suggested to be cited/discussed, such as: Chinese Chemical Letters,2020, 31 (12), 3190-3194;  J. Biomed. Nanotechnol,2019, 15, 1357-1370;ACS Biomaterials Science & Engineering,2019, 5 (10), 5384-5391

5 the language need to be double checked

Author Response

Dear Reviewer,

Thank you for your decision to allow the manuscript revision to meet the Reviewers’ requests. In accordance with the Reviewers’ recommendations, we have performed a revision of the manuscript.

Below, we present the Reviewers’ comments in Italics and our response to them in plain text. All the revisions made in the text of the manuscript are seen green color type submitted as a Review Only Supporting Information.

This study topic is interesting and important. Overall, this report has good quality and the authors have provided some results to support the significance of this study. Reasonable revisions are needed before acceptance.

We thank the Reviewer for the positive estimation and useful comments on our study.

Comments and suggestions:

  1. The reason for choosing/designing this study need to be explained more

The reason for choosing this object of study stems from the general problem of the search of new collagen-based materials with high stiffness and strength, for specific tasks of tissue engineering (tissues in the body which undergo high mechanical loads).

  1. An illustration figure about this study and giant squid mantle is suggested

The study design is presented in the Graphical Abstract. We are not sure we should repeat it in the manuscript body. We added a sketch of the giant squid mantle to the Materials and Methods section (please see Figure 15).

  1. Degradation test is suggested if possible

In part, the material degradation was studied in the experiment on the collagenase digestion. We plan to conduct a more extensive study on the material biodegradation and biocompatibility. Such experiments have been just started to become a part of the next publication.

  1. More background and refs about the medical applications of collagen scaffolds are suggested to be cited/discussed, such as: Chinese Chemical Letters,2020, 31 (12), 3190-3194; J. Biomed. Nanotechnol,2019, 15, 1357-1370; ACS Biomaterials Science & Engineering,2019, 5 (10), 5384-5391

We thank the reviewer for the references, we added them to the Introduction section.

  1. The language need to be double checked

We rechecked the language in the manuscript.

Round 2

Reviewer 1 Report

The authors did not perform any of the experiments suggested, and although I understand that the main scope of the paper was a chemical characterization of the membranes I still deem necessary at least some data on the biosafety of the membranes from a bacterial point of view, which is a starting point for any tissue engineering perspective use. Otherwise it can't be stated that they would be an optimal material for tissue engineering purposes.

The authors should test bacterial adhesion and proliferation by common standard methods allowing bacterial counting (there are many, i.e. live/dead tests, qPCR, colony formation, any of these could be used). Strains commonly used for these tests are E. coli, S. aureus, P. aeruginosa.